# Toll-like Receptor 4 in Acute Kidney Injury

**DOI:** 10.3390/ijms24021415

**Published:** 2023-01-11

**Authors:** Patricia G. Vallés, Andrea Fernanda Gil Lorenzo, Rodrigo D. Garcia, Valeria Cacciamani, María Eugenia Benardon, Valeria Victoria Costantino

**Affiliations:** 1Área de Fisiopatología, Departamento de Patología, Facultad de Ciencias Médicas, Universidad Nacional de Cuyo, Centro Universitario, Mendoza 5500, Argentina; 2IMBECU-CONICET (Instituto de Medicina y Biología Experimental de Cuyo—Consejo Nacional de Investigaciones Científicas y Técnicas), Mendoza 5500, Argentina; 3Área de Biología Celular, Departamento de Morfofisiología, Facultad de Ciencias Médicas, Universidad Nacional de Cuyo, Centro Universitario, Mendoza 5500, Argentina

**Keywords:** Toll-like receptor 4, acute renal injury (AKI), inflammation, ischemia/reperfusion, sepsis-associated AKI, cell regeneration

## Abstract

Acute kidney injury (AKI) is a common and devastating pathologic condition, associated with considerable high morbidity and mortality. Although significant breakthroughs have been made in recent years, to this day no effective pharmacological therapies for its treatment exist. AKI is known to be connected with intrarenal and systemic inflammation. The innate immune system plays an important role as the first defense response mechanism to tissue injury. Toll-like receptor 4 (TLR4) is a well-characterized pattern recognition receptor, and increasing evidence has shown that TLR4 mediated inflammatory response, plays a pivotal role in the pathogenesis of acute kidney injury. Pathogen-associated molecular patterns (PAMPS), which are the conserved microbial motifs, are sensed by these receptors. Endogenous molecules generated during tissue injury, and labeled as damage-associated molecular pattern molecules (DAMPs), also activate pattern recognition receptors, thereby offering an understanding of sterile types of inflammation. Excessive, uncontrolled and/or sustained activation of TLR4, may lead to a chronic inflammatory state. In this review we describe the role of TLR4, its endogenous ligands and activation in the inflammatory response to ischemic/reperfusion-induced AKI and sepsis-associated AKI. The potential regeneration signaling patterns of TLR4 in acute kidney injury, are also discussed.

## 1. Introduction

Acute kidney injury (AKI) is a common clinical state resulting from pathogenic conditions such as ischemic and toxic insults. AKI occurs in up to 35% of hospitalized patients and it is associated with high mortality [1]. The incidence of AKI after general surgery has been reported to be about 1%, whereas the incidence among critically ill patients can be as high as 70%, with an in-hospital mortality of 50% when AKI is part of the multiple organ dysfunction syndrome [2,3]. AKI is an independent risk factor for death [4], and patients who survive have an increased risk to develop chronic kidney disease. Although the involvement of multiple types of cells in the pathophysiology of AKI is becoming increasingly clear, the precise mechanisms for this “AKI to Chronic Kidney Disease (CKD) progression” are still unknown, and no drug has been shown to halt this progression. Therefore, a better comprehension of the physiopathology associated to this syndrome is necessary to identify novel therapeutic approaches.

The etiology and pathophysiology of AKI are complex and multifactorial. The underlying pathophysiological mechanisms include hemodynamic changes, direct tubular toxicity (mainly in proximal tubular cells), obstruction and dysfunction of microvascular vessels, congestion of tubular lumen, and renal inflammation [5]. These pathogenic mechanisms may co-exist in AKI-patients, thus complicating diagnosis and treatment. AKI depends on the duration and severity of the insult [6]. When acute renal damage occurs, there is a first phase of tubular death, followed by a phase of cell regeneration and recovery of the renal function. Massive tubular cell death results from different causes, such as toxic insults, sepsis, oxidative stress, or ischemia, among others [7]. During this first phase, tubular cell death initiates an innate immune response. Damaged or dying cells release endogenous molecules called damage/danger-associated molecular patterns (DAMPs) system in a fashion analogous to pathogen-associated molecular patterns (PAMPs), molecules released by pathogenic bacteria or viruses, activate identical pattern recognition receptors such as Toll-like receptors on tissue-resident cells (dendritic cells, fibroblasts, and tubular cells) or recruited leukocytes [8,9] causing them to secrete proinflammatory cytokines and chemokines [8,10,11].

Toll like receptors (TLRs) are immunity sensors that recognize a wide variety of endogenous and exogenous molecules present in AKI, and promote activation of intracellular pathways associated to renal damage [12].

This review summarizes the present data regarding the role of TLR4 and its endogenous ligand activation in the course and development of acute kidney injury (AKI). The possible regeneration signaling patterns of TLR4 in AKI are also explored.

## 2. TLRS Structure and Ligands

TLRs are type I integral membrane glycoproteins characterized by the extracellular domains containing varying numbers of leucine-rich-repeat (LRR) motifs responsible for ligand recognition and a cytoplasmic signaling domain, homologous to that of the interleukin 1 receptor (IL-1R), termed the Toll/IL-1R homology (TIR), required for the activation of downstream signal pathways domain [13,14].

TLRs are largely divided into two subgroups depending on their cellular localization and respective PAMP (pathogen-associated molecular patterns) ligands. According to their localization, the TLR family can be divided into two subgroups. TLRs that are found on the cell surface membranes and TLRs that reside in endosome membranes. TLR1, TLR2, TLR4, TLR5, TLR6, and TLR10 are expressed on the cell surface membrane, with TLR2 forming heterodimers with TLR1, TLR6 or TLR10. The intracellular TLRs include TLR3, TLR7, TLR8, TLR9, TLR11, TLR12, and TLR13 and they are found localized in intracellular compartments, such as the endoplasmic reticulum, Golgi apparatus, endosomes and lysosomes [14].

TLRs recognize PAMPs (pathogen-associated molecular patterns), which are structural motifs found in viruses, fungi, and bacteria [15]. Cell membrane TLRs (TLR4, TLR5 and TLR10 as well as heterodimers of TLR2 with TLR1 and TLR6) are recruited to phagosomes after activation by their respective ligands e.g., LPS, lipoproteins, flagellin) [16]. By contrast, TLRs involved in the recognition of nucleic acid-like structures e.g., dsRNA, ssRNA, unmethylated CpG motifs are localized in the endoplasmic reticulum, endosomes and lysosomes (TLR3, TLR7 and TLR9) [17].

Furthermore, TLRs can also recognize endogenous stress signal or DAMPs (damage-associated molecular patterns), endogenous factors that are normally sequestered intracellularly and are, therefore, unexposed from recognition by the immune system under normal physiological conditions. Although, under conditions of cellular stress or injury, these molecules can then be released into the extracellular environment by dying cells and trigger inflammation in sterile conditions. The type of cell death notably affects its immunogenicity and ability to release immunostimulatory DAMPs [18]. These last ones are interpreted as signals of a potential danger to the host [19]. Under conditions of extreme damage (for example, ischemia or trauma), necrosis usually occurs when apoptosis fails to happen (sterile inflammation) [18]. A consequence of necrotic cell death is the loss of plasma membrane integrity, allowing the escape of intracellular material from the cell. DAMPs derived from necrotic cells include the chromatin-associated protein high-mobility group box 1 (HMGb1) [20], heat shock proteins (HSPs) [21] and purine metabolites such as ATP [22] and uric acid [23]. In addition to DAMPs from an intracellular source, extracellularly located DAMPs are also released by extracellular matrix degradation during tissue injury. Extracellular matrix fragments, such as hyaluronan, heparan sulphate and biglycan, are generated as a result of proteolysis by enzymes released from dying cells or by proteases activated to promote tissue repair and remodeling [24]. Along with intracellular molecules, intracellular stores of biologically active pro-inflammatory cytokines and chemokines such as, IL-1α [25] and IL-33 [26], may be released by necrotic cells. Although these factors are not conventionally considered as DAMPs, they can mediate sterile inflammatory responses [18].

TLR activation leads to the initiation of intracellular signaling pathways that elicit the expression of inflammatory genes, such as, cytokine, essential for host defense, and interferons (type I IFNs), critical for antiviral defense [27].

## 3. TLR4 Expression

The first identified member of mammalian TLR family was TLR4 [28]. Although many TLRs have been described, TLR4 is so far the most extensively studied. TLR4 is extensively expressed in resident immune and in renal parenchymal cells at physiological conditions.

In the kidney, the tubular epithelial cells and mesangial cells express TLR1 through TLR4 and TLR6 [29]. TLR4 expression is elevated in renal cortex, whereas low levels have been described in renal medulla. In the cortex, TLR4 expression is mainly detected in proximal and distal tubules, but also in podocytes, glomerular mesangial cells, peritubular endothelial cells and collecting duct cells [30]. In normal conditions, renal TLR4 expression is low, however, the expression of this molecule increases in response to renal injury and/or infection.

## 4. The Intracellular Signal Transduction of TLR4

Toll-like receptor 4 serves as a major receptor mediating the effects of Lypopolysccharide (LPS), an integral unit of the gram-negative bacterial cell wall, and has been implicated in the pathogenesis of AKI [31,32]. In a cell, TLR4 is found both on the cell surface and in intracellular phagosomes where TLR4 forms a receptor complex with myeloid differentiation factor 2 (MD2) [33]. The role of MD2 in LPS signaling is further supported by an animal study, where MD2 deficient mice remained unresponsive to gram-negative bacterial LPS, which usually mediate endotoxemic shock [34].

Binding interaction of LPS and TLR4 induces dimerization of the receptor. Two symmetrically arranged copies of TLR4–MD2–LPS complex represent the TLR4 dimer [35]. TLR4 dimerization is followed by its interaction with several intracellular adaptor proteins. Cluster of differentiation 14 (CD14), a co-receptor, transfers LPS to the TLR4/MD2 complex, as well as facilitates endocytosis of TLR4 [36,37]. LPS upregulates the production of pro-inflammatory mediators via MyD88-dependent pathway and TRIF-dependent (MyD88-independent pathway), which signal from the cell surface and endosomes respectively [38].

Individual TLRs differentially recruit members of a set of cytosolic TIR domain containing adaptors such as (1) myeloid differentiation factor 88 (MyD88); (2) MyD88 adaptor-like (MAL, also known as TIRAP); (3) TIR domain-containing adaptor protein-inducing IFN-β (TRIF, also known as TICAM1); and (4) TRIF-related adaptor molecule (TRAM, also known as TICAM2) [14].

TLRs signaling pathways can be largely classified as either MyD88-dependent pathway, which drives the induction of inflammatory cytokines, or TRIF-dependent pathway or MyD88-independent pathway, which is responsible for the induction of type I interferon as well as inflammatory cytokines [14,39]. TLR4 is the only TLR that uses all four adaptors and activates both the MyD88 and TRIF-dependent pathways (Figure 1). TLR4 initially recruits TIRAP at the plasma membrane, and subsequently, facilitates the recruitment of MyD88 to trigger the initial activation of NF-κB and MAPK [40]. MyD88-dependent pathway requires the formation of myddosome through the assembly of proteins like MyD88, IRAK2, and IRAK4. Formation of myddosome results in the activation of TRAF6 and TAK1 complex [41]. TAK1 further facilitates the kinases (IKK) dependent phosphorylation and degradation of Iκβ (inhibitor of NF-κB), enabling the nuclear translocation of NF-κB, which initiates the transcriptional synthesis of proinflammatory cytokines followed by activation of NLRP3 and release of IL-1β and IL-18 [42,43].

Additionally, TAK1 also phosphorylates MAPKs, which account for the activation of another transcription factor known as activating protein-1 (AP-1) [44]. The TRIF-dependent pathway or MyD88 independent pathway, culminates in the activation of both IRF3, a member of the interferon regulatory transcription factor (IRF) family, and NF-κB [44,45]. The MyD88-independent pathway intervenes through the endocytosis of activated TLR4 dimer and subsequent recruitment of adaptor proteins like TRAM and TRIF to endosomal TLR4. TRAM is considered as a bridging adaptor between TLR4 and TRIF. TRIF pathway stimulates the transcription of type I interferons (IFNs) dependent on TRIF mediated activation and nuclear translocation of IRF3. TRIF functions as an activator of TRAF3, which further phosphorylates IRF-3 through the recruitment of TANK (TRAF associated NF-κB activator), TBK1 (TANK binding kinase-1) and IKKε (IκB kinase ε) complex, resulting in the transcriptional activation of gene encoding type I interferon [46]. The TRIF-TRAM dimer is also capable of stimulating TRAF6 which then activates downstream MAPK and NF-κB as described in MyD88-dependent pathway [47]. Thus, TLR4 activates the MyD88-dependent pathway earlier than the TRIF-dependent pathway [33].

### 4.1. TLR4 Signaling in Ischemia/Reperfusion-Induced AKI

Ischemic acute renal injury remains an area of enormous clinical importance and cost. Advances in understanding the cellular and molecular aspects have been made, however, little progress has been achieved in the translation of these findings to the clinical area. Ischemia/reperfusion (I/R) injury can be described as tissue destruction that happens when blood supply goes back to the tissue after a period of ischemia [48]. The tissue damage associated with I/R is mainly linked with trauma, stroke, myocardial infarction and solid organ transplantation. Though the cause of this injury is multifactorial, data suggests that the innate immune system has a significant role in initiating the inflammatory cascade leading to deleterious changes in kidney tissues [49]. A robust inflammatory response, triggered by hypoxia and by the process of reperfusion, determine the outcome of the ischemic organ.

First of all, tubular cell death initiates an innate immune response. Necrotic cells release intracellular molecules, called damage-associated molecular patterns (DAMPs), activate identical pattern recognition receptors such as Toll-like receptors on tissue-resident cells (dendritic cells, fibroblasts, and tubular cells) or recruited leukocytes [8,9,50]. These responses further conduct various types of inflammatory cells to enter the site of cell deaths, leading to additional cell deaths, forming a vicious cycle of cell death/inflammation [51,52]. The extent of cell death determines the extent of AKI, and thereby, minimizing injury itself is one of the important approaches to prevent severe AKI. This DAMP-associated sterile inflammation has been considered one of the earliest processes following injury and innate immunity cell-derived cytokines influence the activation of the adaptive immune response, which also enhances and sustains these responses. During renal ischemia-reperfusion injury, upregulation of TLR4 is found in tubular epithelial cells, vascular endothelial cells and infiltrating leukocytes. The TLR2 and TLR4 are at the center of these inflammatory actions of I/R-induced AKI [53,54,55]. In the kidney, the majority of the constitutive expression of TLR4 (and TLR2) is found in glomerular endothelial cells, in podocytes and renal tubular epithelial cells. During reperfusion, changes in TLR4 expression may occur non-concurrently on all kidney cell types. The expression is increased both on endothelial and renal tubular epithelial cells following injury. Endothelia of the vasa recta of the inner stripe of the outer medulla exhibited TLR4 upregulation after 4 h following reperfusion compared with 24 h for renal tubular epithelial cells [56]. After TLR4 stimulation, the main upregulated cytokines are IL-6, IL-1β and TNF-α. This is accompanied by increased expression of macrophage inflammatory protein-2 (MIP-2) and monocyte chemoattractant protein-1 (MCP-1). Both are involved in the recruitment of monocytes [57,58].

Tubular cell necrosis and apoptosis are major factors in the regulation of inflammation, since dying cells release intracellular molecules which are referred as DAMPs. DAMPs activate a set of pattern recognition receptors such as TLRs on tissue-resident cells. There is strong evidence that TLR2 and TLR4 participate in this process [59,60]. Activation of TLR4 promotes the release of proinflammatory mediators, facilitates leukocyte migration and infiltration, activates the innate and adaptive immune system and potentiates renal fibrosis.

Kim et al. were the first to show the gene and protein expression upregulation of TLR2 and TLR4 in rat kidney tissues after reperfusion in ischemic injury [60]. Their results of in situ hybridization and RT-PCR of TLR2 and TLR4 mRNA expression in I/R injury kidneys showed, a strong signal in cells from proximal tubule, outer strip of the outer medulla (OSOM) and in thick ascending limbs (TAL) of the inner strip of the outer medulla ISOM—At day 3 that it peaked at day 5. Likewise, Wolfs et al. demonstrated that the expression of TLR2 and TLR4, constitutively expressed in both proximal and distal tubules, the thin limb of the Henle loop and in the collecting ducts, was upregulated in these areas at day 5 after I/R. Increased expression on TLR2 and TLR4 expression up to four to five fold over basal levels was shown especially in, the distal tubular epithelium, the thin limb of Henle’s loop, and collecting ducts [61].

The individual role of TLR4 on kidney I/R injury was described by Pulskens et al. [62], demonstrating the proinflammatory role of TLR4, as shown by a reduction of infiltrating granulocytes and chemokines in kidneys of TLR4^(−/−)^ mice compared with wild-type mice. Since an exaggerated inflammatory response could lead to more severe tissue damage, this increased inflammation could explain why TLR4^(−/−)^ mice showed less tubular injury and a more preserved renal function, compared with wild-type mice upon kidney I/R injury. Epithelial and leukocyte-associated functional TLR4 contribute in a similar proportion to renal dysfunction and injury, as assessed by bone marrow chimeric mice. No significant differences were found in renal function and inflammation in both MyD882/2 and TRIF-mutant mice when compared to their wild-types, allowing the identification of TLR4 as a cellular sentinel for acute renal damage. These last results were corroborated by studies from Shigeoka et al. [63] in TLR2 mice deficient in TLR2, MyD88, TRIF, and MyD88 plus TRIF. The absence of TLR2, MyD88, and MyD88 plus TRIF conferred both physiologic and histologic protection against sublethal ischemia at 24 h. Interestingly, TLR2-deficient mice were better protected from ischemic renal injury than those deficient for the adapter protein MyD88, raising the intriguing possibility that TLR2-dependent/MyD88-independent pathways also contribute to kidney injury. In contrast, Wu et al., in bone marrow chimeric mice studies demonstrated that TLR4 signaling through the MyD88-dependent pathway was required for the full development of kidney I/R injury, as both TLR4^(−/−)^ and MyD88^(−/−)^ mice were protected against kidney dysfunction, tubular damage, neutrophil and macrophage accumulation, and proinflammatory cytokines release [9]. Several renal endogenous ligand expressions increased after I/R injury, including biglycan, HMGB1 and hyaluronan, providing evidence that one or more of these ligands may be the source of TLR4 activation. Chen et al. confirmed the role of TLR4 in ischemic kidney injury using mice harboring spontaneous disabling mutations of the receptor and generated chimeras between TLR4^(−/−)^ and TLR4^(+/+)^ mice. The sequence of events proposed, ascribes to the release of HMGB1 from damaged epithelial and endothelial cells. The HMGB1 role, as a trigger for TLR4, induces activation of leukocytes and macrophages which subsequently secretes proinflammatory IL-6 in TLR4^(+/+)^ mice. Thus, after injured kidney infiltration leukocytes produce IL6 when their TLR4 receptors interact with HMGB1 released by injured renal cells [64]. Interestingly, the treatment with an antibody against HMGB1 reduced recruitment of neutrophils and macrophages and diminished inflammation and apoptosis in tubular epithelial cells from TLR4^(+/+)^ mice, but not in TLR4^(−/−)^ mice, indicating the key role of HMGB1/TLR4 axis in I/R injury [65].

The in vivo experiments conducted by Rusai et al. [66] showed non-synergistic beneficial effects in TLR2 and TLR4 double knockout mice submitted to I/R as compared to single gene deletion. No difference was shown in the number of apoptotic tubular cells and in the nuclear translocation of NF-κB between the TLR-gene-targeted groups. In addition, in vitro experiments did not demonstrate an additional effect of the double genetic deletion compared with the single gene deletion related to tumor necrosis factor (TNF-α) and interleukin IL-8 production, in hypoxic isolated proximal tubular epithelial cells. Protection from microvascular rarefaction but not from the development of fibrosis was demonstrated in kidney I/R injury induction in TLR4 knockout mice [67].

A recent study identified miR-27a as a negative regulator of TLR4 and demonstrated that overexpression of this miRNA reduced the expression of TLR4 by binding to the 3′UTR of TLR4 and consequently diminished I/R-mediated renal inflammation, cell adhesion and cell death [68].

Kidney transplantation is a pathological situation closely related to I/R damage. Increasing evidence has shown that DAMPs generation during I/R activates TLR4 and TLR2 and this activation results in the production of proinflammatory cytokines and chemokines, facilitating leukocyte migration and infiltration [48].

A significant induction in TLR4 expression restricted to the glomerular compartment, has been demonstrated in acute-rejecting allografts accompanied by a significant increase in CC chemokine expression within the graft, as well as in urinary CC chemokine excretion [69]. Likewise, previous studies have reported significantly lower levels of proinflammatory mediators and cellular infiltration associated with preserved graft function in TLR4-knockout and TLR2-knockout mice compared to wild-type mice [70]. Kaczorowski et al. showed that TLR4 signaling is dominant in both, the systemic and intragraft inflammatory responses that occur after cold ischemia reperfusion in the setting of organ transplantation [71]. In human kidney transplants, the roles of TLR2 and TLR4 signaling, have also been demonstrated in acute allograft rejection. Palmer et al. suggested that the activation of innate immunity through TLR4 in a donor kidney contributes to the development of acute rejection after renal transplantation [72]. Krüger et al. looked at the expression levels of TLR4. Whilst TLR4 is constitutively expressed in donor organs, the level of expression was significantly higher in non-heart beating donor kidneys which also correlated with increased levels of HMGB-1. They also genotyped the organs for known TLR4 loss of function mutations which alter signaling in response to HMGB-1 and other ligands. Those organs carrying TLR4 mutations exhibited reduced levels of cytokines and a higher rate of immediate graft function [73]. Inhibition of TLR4 could diminish ischemia-reperfusion injury as well as delayed graft function and allograft rejection [55]. Even though, it is not enough to conclude that excessive TLR4 signaling is an independent risk factor for acute rejection (AR), early signaling through TLR4 may represent a rate-limiting step in the rejection cascade. These recent findings provide an excellent platform for discussing the complexity of danger signaling in the development of ischemic/reperfusion-induced AKI [41].

### 4.2. TLR4 Signaling in Sepsis-Associated AKI

Sepsis-associated AKI is a life-threatening complication that is associated with high morbidity and mortality in patients who are critically ill. Early supportive interventions in sepsis reduce mortality, although it is less clear that they prevent or ameliorate sepsis-associated AKI, since specific mechanisms underlying AKI attributable to sepsis are not fully understood. The fundamental mechanisms of sepsis-associated AKI include microcirculatory dysfunction, inflammation and adaptive bioenergetic, and metabolic downregulation in renal tubules [74]. After bacterial infection, a hyperactive dysregulated innate immune response leads to a cascade of proinflammatory molecules, contributing to sepsis-associated AKI by the activation of cellular innate immunity and the complement system. Cell death pathways and immune cells are activated, leading to infiltration of T cells, macrophages, neutrophils and amplification of tissue injury [75,76,77].

Early events trigger innate immunity through the host response to danger-associated molecular patterns (DAMPs), normally endogenous “hidden” intracellular molecules that are released by dying or damaged cells or pathogen-associated molecular patterns, small molecular motifs including LPS, flagellin, double-stranded RNA, and CpG DNA, which are ligands for pattern recognition receptors, such as TLRs, NOD-like receptors (NLRs), and RIG-I–like receptors [78]. Early deaths are a result of a hyperinflammatory “cytokine storm”. The role of the inflammatory response in sepsis-associated AKI points strongly towards TLR4 as a potential mediator in the development of sepsis-associated AKI. LPS is the main ligand for TLR4 [79]. The interaction between LPS and both systemic and renal TLR4 has been reported in sepsis-associated AKI [80]. Recent animal studies have demonstrated that targeting TLR4 with specific antibodies reduce endotoxemia-associated mortality [81,82]. In humans, TLR4 polymorphisms have been related with a reduced LPS-mediated inflammatory response [83]. In the context of renal damage, TLR4 can be activated by alarmins (endogenous ligands for TLR4; i.e., HGMB1 and Hsp70) as a consequence of cellular stress, amplifying kidney injury [84,85].

In vitro stimulation of human tubular epithelial cells with HMGB1 confirmed that this molecule can stimulate proinflammatory responses through TLR4. In a human study, Krüger, et al. demonstrate that TLR4 is constitutively expressed in kidneys and that tubule cells in damaged kidneys also stain positively for HMGB1, a known endogenous TLR4 ligand [73]. Leukocytes leaving peritubular capillaries have a close proximity to tubular epithelial cells and can directly activate these cells by releasing pro-inflammatory mediators and DAMPs. Based on the special location of tubular epithelial cells, they can also be activated from the tubular side [80]. DAMPs, PAMPs and pro-inflammatory cytokines are filtered in the glomerulus, enter the proximal tubule and can directly activate tubular epithelial cells resulting in a change of the metabolic and functional state of these cells. It has been recently shown that these molecules can activate tubular cells by binding to TLR2 and TLR4 [86,87].

During sepsis, endotoxin is readily filtered and internalized by S1 proximal tubule cells through local TLR4 receptors and through fluid-phase endocytosis. Only receptor-mediated interactions between endotoxin and S1 caused oxidative stress in neighboring S2 tubule cells, suggesting that targeting TLR4 signaling may have value in preventing or treating AKI [86]. A study performed on C3H/HeJ mice (TLR4 mutant) and C3H/HeOuJ control mice revealed that LPS acts on extrarenal TLR4, thereby leading to systemic TNF-α release and subsequent acute renal injury. Besides, TNF-α release, renal apoptosis and neutrophil migration were significantly low in C3H/HeJ mice when compared with C3H/HeOuJ control mice [31]. Another animal study suggests that sepsis-induced TLR4 activation upregulates neutrophil infiltration and the expression of proinflammatory cytokines in tubular epithelial cells, resulting in the development of acute kidney injury. Knockout of the TLR4 gene in C57BL/6 mice afford protection against sepsis-associated AKI [88].

TLR4 expression in the thick ascending limb is increased in response to sepsis and I/R injury, and this segment has been implicated in mediating inflammatory renal injury during these conditions [27,89]. Moreover, the medullary thick ascending limb (MTAL) has been identified as a site of cell damage and tubule dysfunction in response to microbial infection. HCO_3_^−^ reabsorption is inhibited by LPS in the MTAL from either the basolateral or luminal cell surface through the activation of TLR4. Although, the underlying signaling mechanisms are different, HCO_3_^−^ reabsorption through the activation of an extracellular-signal-regulated kinase (ERK) pathway, is decreased in the presence of basolateral LPS [90]. In addition, basolateral LPS inhibits HCO_3_^−^ absorption in the MTAL through activation of a TLR4/MyD88/mitogen-activated protein kinase (MEK/ERK) pathway coupled to inhibition of sodium–hydrogen exchanger (NHE3) [91]. NHE3 has been identified as a target of TLR4 signaling. Thus, bacterial molecules can impair the absorptive functions of renal tubules through the inhibition of the exchanger NHE3 [92].

The ERK pathway links TLR4 to downstream modulation of ion transport proteins. Recent work suggests that one important mechanism underpinning septic AKI is that this inflammatory pathway induces renal tubular transport dysfunction, which enhances NaCl delivery to the macula densa and increases tubule-glomerular feedback therefore decreasing glomerular filtration rate.

In addition to inhibition by gram-negative LPS through TLR4, reabsorption of HCO_3_^−^ by the MTAL is inhibited by gram-positive bacterial molecules through TLR2. The TLR2 receptor is expressed selectively in the basolateral membrane of MTAL cells, in contrast to TLR4, which is expressed in both basolateral and luminal membrane domains [90,93]. The MTAL HCO_3_^−^ reabsorption is inhibited by bacterial lipopeptides and gram-positive bacterial cell wall structures (lipoteichoic acid and peptidoglycan) recognized by TLR2 [93].

Moreover, during sepsis, gram-positive bacterial molecules acting through TLR2 and gram-negative LPS acting through TLR4, can function through parallel signaling pathways to impair MTAL transport. These findings have important implications for the pathogenesis of kidney dysfunction during polymicrobial sepsis, because they show that gram-negative and gram-positive bacterial molecules can act independently and additively to impair renal tubule function by activating intracellular signals through TLR4.

In the presence of soluble CD14 and LPS binding protein (LBP), proximal tubular epithelial cells have been demonstrated to be more sensitive to LPS activation, resulting in cytokine production. The presence of soluble CD14 and LBP was required for cytotoxicity induced by low concentrations of LPS in proximal tubular epithelial cells. Human proximal tubular epithelial cell death due to sepsis/inflammation showed characteristics of both necrosis and apoptosis, where necrosis has been primarily attributable to LPS stimulation, while soluble CD14 has been required for induction of renal cell apoptosis [94].

In addition to PAMPS and DAMPS that bind to TLRs on the cell surface, the cytoplasmic NOD-like receptor protein 3 (NLRP3) inflammasome is integrated to the inflammatory cascade in sepsis-associated AKI [95,96]. Whereas detection of extracellular LPS and ensuing immune responses through TLR4 signaling pathway plays a major role in the primary detection of LPS, the recognition of cytosolic LPS by intracellular proteases caspase-4/5 (and their mouse homologue caspase-11), leads to the activation of NLRP3 inflammasome [97]. NLRP3 inflammasome activation, triggers the secretion of IL-1b and IL-18 via caspase-1 mediated processing of pro-IL-1b and pro-IL-18, and pyroptosis accounting for endotoxin-related pathology [18,97]. In a mouse model of sepsis, NLRP3 knockout reduced caspase-1 and IL-1b/IL-18 levels in the kidney and attenuated kidney injury [98]. Recent investigations have shown that cellular energy metabolism, especially in ATP metabolism, is pivotal for regulating the NLRP3 inflammasome [99,100].

Apart from inflammation, sepsis-associated AKI is characterized by additional pathological mechanisms: dysregulated renal microcirculation and metabolic adaptation of renal cells to injury. Although the connection with TLR4 might not be direct, as in the case of the inflammatory response, recent evidence suggests a potential beneficial effect of TLR4 inhibition against these harmful processes. Microvascular function is a major contributor to the microcirculation [101] and despite maintenance of sustained renal blood flow, altered microcirculatory flow may lead to altered kidney tissue hypoperfusion. By using intravital video microscopy Wu et al. demonstrated that peritubular capillary flow decreased by 88% in LPS-induced sepsis [102]. By multiparametric photo-acoustic microscopy Sun et al. showed that sepsis induced an acute and significant reduction in peritubular capillary oxygen saturation of hemoglobin, concomitant with a marked reduction in kidney ATP levels [103]. These intravital studies demonstrated that sepsis altered microvascular flow, resulting in tissue hypoxia and compromised bioenergetics [102,103]. The microvascular dysfunction is heterogeneous and relates to patchy areas of leukostasis, with associated changes in oxygen extraction (metabolic rate of oxygen). These studies showed local microvascular dysfunction and patchy areas of oxygen saturation were likely related to endothelial dysfunction and inflammatory and other oxidative factors [103,104]. The inflamed local microenvironment and its contribution to microvascular dysfunction during sepsis has been widely reported [103,104,105]. Endothelial dysfunction in sepsis is characterized by dysregulated renal blood flow and reduced glomerular filtration rate [106]. Pharmacological inhibition or genetic deletion of TLR4 in pre-clinical sepsis models has been associated with reduced glomerular endothelial swelling and vascular permeability, respectively [88,107].

Pericytes, which intermittently surround capillary endothelial cells to maintain microvascular homeostasis including blood flow and vascular permeability, are considered the main source of myofibroblasts in chronic kidney disease with progressive fibrosis [108,109]. Recent studies suggest that pericytes could be implicated in the heterogeneity of microcirculatory abnormalities during the progression of sepsis-associated AKI, as supported by recently identified molecular mechanisms underlying sepsis-associated dysfunction of kidney pericytes [105,110,111]. In a swine model of LPS-induced AKI, pericyte-to-myofibroblast transdifferentiation (PMT) was observed within 9 h after LPS challenge, as evaluated by the reduction of physiologic platelet-derived growth factor receptor beta (PDGFRβ) expression and the dysfunctional α-SMA increase in peritubular pericyte. This pericytes events might be mediated by the secretion of LPS-binding protein (LBP) and subsequent stimulation of TLR4 signaling in pericytes. Moreover, LPS-stimulated pericytes secreted LBP and TGF-β and underwent PMT also upon TGF-β receptor-blocking, indicating the crucial profibrotic role of TLR4 signaling [112].

The Shiga toxin (Stx)-producing *Escherichia coli*-associated hemolytic uremic syndrome (STEC-HUS) with diarrhea at presentation, is the most frequent cause of AKI in infants and young children in Argentina [113]. Endothelial injury has been recognized as a trigger event in the microangiopathic process [114]. Host endothelial cell inflammatory response to *E. coli* Stx and/or LPS contributes to the ongoing vascular damage from the infection, which results in HUS [115]. LPS augments Stx toxicity [116]. Activation of a strong inflammatory response causing the release of cytokines and chemokines, the recruitment of leukocytes and monocytes, and the activation of complement and thrombotic cascades have been shown during the early stage of STEC-HUS [117]. In a retrospective study of critically ill HUS pediatric patients with evidence of *E. coli* infection conducted through a period of 15 years, we described multiorgan involvement at the initial phase of disease. Severe clinical outcome at onset suggests an amplified inflammatory response after exposure to Shiga toxin and/or *E. coli* LPS. Patients who suffered from STEC-HUS associated with severe neurological involvement, hemodynamic instability and AKI require intensive care and focused therapy [118]. In a clinical study we investigated TLR4 surface receptor expression on peripheral blood neutrophils and their ability to modulate inflammatory cytokine release, in pediatric patients who suffered from Hemolytic Uremic Syndrome. Isolated leucocytes from the STEC-HUS-onset patients exhibited significantly higher mRNA TLR4 expression than the controls. Moreover, TLR4 protein expression on neutrophils determined by flow cytometry was upregulated, driving dependent proinflammatory cytokine, tumor necrosis factor alpha, IL-8 increase, and decreased anti-inflammatory IL-10 release at HUS onset compared with patients with enterohemorrhagic *Escherichia coli* (EHEC) diarrhea, and healthy children. Conversely, significant reduction of the neutrophil TLR4 receptor expression and lack of cytokine-responsive element activation, was shown in patients 3 and 10 days after the STEC-HUS onset. These results suggest that TLR4 expression may be differently regulated on neutrophils. They could be dynamically modulated across the early development of STEC-HUS on neutrophils, resulting in the negative regulation preceded by TLR4 overactivation [119].

The receptor-ligand intracellular membrane trafficking and routing is crucial to reach a controlled inflammatory response and a timely and regulated termination of inflammatory signaling. Modulation of TLR4 activation and their downstream-related signaling pathways include several mechanisms, such as soluble decoy receptors, transmembrane regulators, cellular trafficking, destabilization of adaptor proteins, ubiquitination, dephosphorylation, transcriptional regulation and feedback inhibition [120].

Rab7b may negatively regulate TLR4 signaling, which promotes TLR4 targeting to lysosomes for degradation [121]. Afterwards, we investigated the role of Rab7b in LPS-initiated TLR4 signaling in monocytes from pediatric patients during the acute course of Shiga toxin-associated HUS. We found that at the very beginning of the disease, days 1 and 4, Rab7b colocalizes with TLR4 in intracellular vesicles with maximal colocalization at day 4. These results suggest a localization of TLR4 within Rab7b-coated vesicles (Figure 2). Both proteins run together in intracellular trafficking directed towards degradation. It is likely that the higher TLR4 observed at the surface level by flow cytometry could demonstrate the recycling of the receptor in a step prior to the proteolytic degradation; and/or, it may be possible that the sustained stimulus of the pathogen could generate de-novo synthesis, which would also contribute to the increased receptor expression in the cell membrane on day 4. As the disease progresses, surface TLR4 decreases as well as the monocyte intracellular cytokine inflammatory response. Monocyte cell analysis on day 10 showed surface TLR4 expression and intracellular proinflammatory cytokines expression near controls. Lower expression of Rab7b and TLR4 and minimal and punctual colocalization of both proteins was observed with no differences related to control (Figure 2). In this way, the early colocalization of Rab7b/TLR4 may account for an adequate/successful trafficking of TLR4 to the degradation pathway leading to the downregulation of proinflammatory state or inflammation during the early follow up (10 days) of the disease. Our findings suggest that Rab7b participates as a negative regulator in the control of the TLR4 endocytic pathway in pediatric STEC-HUS patient monocytes (Figure 3). The resulting decline in monocyte cell cytokine generation, is demonstrated by the induction of the TLR4 receptor endocytosis during the early follow up of STEC-HUS [122].

Hence, strict negative regulation of TLR4 signaling is required to protect the host from an exacerbated inflammatory response.

### 4.3. TLR4 Inhibition in Acute Kidney Injury

The effect of anti-inflammatory agents targeting TLR4 have been analyzed in preclinical studies during AKI [41,123]. They are low-molecular weight compounds of natural and synthetic origin that can be considered leads for drug development. Given the role of TLR4 in the pathogenesis of AKI, several therapeutic modulators have been devised to regulate TLR4 expression These compounds can be categorized as antibodies, small-molecule inhibitors, peptides, microRNAs, nanoparticles, lipid A analogs, and derivatives of natural products [124,125,126,127]. As previously reported, some natural (loganetin, resveratrol and curcumin) and synthetic compounds (TAK-242, Eritoran and hydrogen sulfide) showed beneficial effects in experimental models of AKI [120]. Several TLR4 antagonists have been synthesized, the majority of them being mimetics of lipid A, the natural MD-2 ligand [128].

Only few synthetized compounds, are currently being evaluated in clinical trials. The best known lipid A mimetic is Eritoran. The Eritoran/MD-2 complex revealed that it binds MD-2 more similarly than lipid A, by accommodating the four fatty acid chains into MD-2 binding pocket. Hence, Eritoran acts as a classic competitive inhibitor of MD-2 competing with LPS for the binding of the MD-2 pocket [129]. After the successful results obtained in animal AKI models, Eritoran was suggested for testing on humans [130,131,132].

A small-molecule compound known as TAK-242 (Resatorvid), a TLR4 antagonists with a chemical structure unrelated to lipid A, selectively inhibits TLR4 signaling by binding directly to a Cys747 in the intracellular TLR4 domain. This binding disrupts the interactions of TLR4 with its adaptor molecules, TIRAP and TRAM [133]. TAK-242 has been admitted to clinical trials.

Even though promising results were observed on animal models, so far no data from clinical trials targeting TLR4 have led to a significant improvement in patients with AKI. Further studies in human beings are necessary to confirm the probably positive effect of TLR4 inhibition in AKI.

### 4.4. TLR4 and Cell Regeneration

The kidney repair process entails interaction of complex events. Reassembly of the actin cytoskeleton, and repolarization of the surface membranes initiate the recovery of proximal tubular cells. The replacement of injured tubular epithelium with functional tubular cells happens in a restricted manner, as the kidney has limited capacity to undergo endogenous tissue remodeling. During renal development, the mesenchymal to epithelial transition (MET) process is controlled by different growth factors e.g., hepatocyte growth factor (HGF) and bone morphogenetic protein-7 (BMP-7). Therefore, a transition of the metanephric mesenchymal cells into polarized epithelial cells may be required in kidney regeneration [134].

Tubular cell injury peaks 2–3 days after injury and is followed by a repair phase in which proliferation of surviving tubular cells promotes tubule regeneration. In the ischemic injury, the inflammatory response is accompanied by the rapid influx of polymorphonuclear leukocytes, lymphocytes, and macrophages into the interstitium. Mononuclear phagocyte/macrophages are one of the major cell types that accumulates around tubules [135]. Mononuclear inflammatory cells have been found in the vasa recta of the outer medulla. Macrophages exhibit a range of phenotypes; this phenomenon has been described as macrophage polarization or heterogeneity. It is the effector phenotype of the recruited macrophages rather than their presence alone, that determines the extent of renal parenchymal injury, as suggested by recent studies of other forms of immune-mediated renal injury. It has been proposed that inflammatory cells play a negative role. Interventions that could mitigate the initial inflammatory response by inhibition of the cytokine action, prevention of inflammatory cell homing, or depletion inflammatory cells, seem to decrease the degree of morphologic and functional injury in renal I/R [136].

To limit overshooting immunopathology in sterile tissue injuries and allow tissue recovery, a number of counter regulatory mechanisms exist that mostly limit immune activation of intrarenal dendritic cells [137]. The phenotypic switching of intrarenal mononuclear phagocytes away from classically activated (proinflammatory) to alternatively activated (anti-inflammatory/proregeneratory) cells is necessary for recovery on AKI. Although, surviving tubular epithelial cells (TECs) enter the cell cycle within few hours on injury, a functional tubular recovery does not occur before the resolution of sterile inflammation is resolved and the tubulointerstitial microenvironment become dominated by proregeneratory factors [138]. These factors are provided in a paracrine manner by other surviving tubular epithelial cells, intratubular progenitor cells, or bone marrow-derived stem cells [139]. The process of repair begins with a marked increase in tubular cell proliferation that peaks on day 3 and slowly declines over the ensuing week. Most dividing cells are tubular cells, with approximately 88% of the bromodeoxyuridine-positive proliferating cells expressing the proximal tubule marker megalin and 1% expressing the thick ascending limb marker Tamm–Horsfall protein. The remaining 10% was unclassifiable. Although dendritic cells and other immune cells play a dominant role in orchestrating the early injury phase of AKI, little is known about the contribution of intrarenal immune cells to the subsequent phase of kidney regeneration [139,140].

The activation of TLRs has also been implicated in epithelial repair by regulating clearance of cellular debris and initiating tissue-repair programs. Inhibin A and decorin belong to the TGF-β signaling pathway. Evidence exists for an antifibrotic activity of soluble decorin, directly interacting with another member of TGF-β superfamily, the connective tissue growth factor (CTGF), and inhibiting apoptosis of renal tubular epithelial cells via the insulin-like growth factor 1 (IGF-1) receptor/Akt-signaling pathway [10]. It has been observed that certain DAMPs activate TLR2 on CD133^+^/CD24^+^ renal progenitor cells and accelerate tubular repair through the release of soluble factors inhibin A and decorin. Blocking of TLRs2 completely abrogate this regenerative effect [141].

Kulkarni et al. identified yet unknown proregeneratory properties of interleukins (ILs), a family of leukocyte-derived mediators on tubular epithelial cells regeneration in AKI. Their results revealed that IL-22 secretion was selectively induced by TLR4 agonists released from necrotic tubular cells, which in turn activated its receptor (exclusively present on tubular epithelia) to accelerate re-epithelization in vivo through activation of the Jak/STAT3 and ERK1/2 pathways. Interstitial mononuclear cells, such as dendritic cells and macrophages, were the predominant source of IL-22 secretion, whereas IL-22 receptor was expressed by tubular epithelial cells exclusively. Neutralization of IL-22 in the healing phase of AKI (2–5 days after injury) by injecting anti–IL-22 antibody significantly impaired tubular recovery. This study showed a role of TLR4 signaling not only for renal immunopathology but also for kidney regeneration in vivo [142].

Nechemia-Arbely et al. described how Renal IL-6 expression and STAT3 activation in renal tubular epithelial cells significantly increased during the development of injury, suggesting active IL-6 signaling. Notwithstanding a loss of renal IL-6 receptors (IL-6R) precludes the activation of classical signaling pathways, IL-6 can stimulate target cells together with a soluble form of the IL-6R (sIL-6R) in a process termed trans-signaling. During AKI, serum sIL-6R levels increased three-fold, leading to a possible role for IL-6 trans-signaling in injury. This signaling reduced lipid peroxidation after injury, suggesting that IL-6 simultaneously promotes an injurious inflammatory response and, through a mechanism of trans-signaling, protects the kidney from oxidative stress [143]. A study on primary human peripheral blood mononuclear cells (PBMCs) and on the monocytic cell line THP-1 showed that activation of Toll-like receptor 2 (TLR2) induced expression and secretion of IL-6 and the generation of sIL-6R [144]. However the same role in the context of AKI, has not been demonstrated until now.

Macrophages exhibit a range of phenotypes. Phenotypic switch of pro-inflammatory macrophages (M1) to anti-inflammatory/pro-regeneratory macrophage cells (M2) is essential for recovery on AKI. Proinflammatory (M1) macrophages are recruited into the kidney in the first 48 h after ischemia/reperfusion injury, whereas, noninflammatory (M2) macrophages predominate at later time points. The “classically” activated (M1) macrophage is induced by exposure to interferonγ (IFNγ) and expresses proinflammatory cytokines. IL-4 stimulated macrophages with an M2 phenotype, but not IFN-γ-stimulated pro-inflammatory macrophages. The IL-4 stimulated M2 marcophages promoted renal tubular cell proliferation 3–5 days after injury. Depletion of macrophages before ischemia/reperfusion diminishes kidney injury, whereas depletion 3–5 days after injury reduce tubular cell proliferation and repair [145].

Biglycan, a small leucine-rich proteoglycan acts as a danger signal and is classically thought to promote macrophage recruitment via Toll-like receptors (TLR) 2 and 4. Roedig et al. showed that soluble biglycan signaling through TLR 2/4 and the CD14 co-receptor regulates inflammation. CD14 was crucial for biglycan-induced renal M1 macrophage polarization in AKI [146].

In a recent study Poluzzi et al. suggest that TLR2/4 co-receptors may determine whether biglycan-TLR signaling is pro-or anti-inflammatory. The authors demonstrated that biglycan proteoglycan promotes macrophages autophagy mechanism by CD44 and TLR4 signaling axis in renal I/R injury and evokes anti-inflammatory response. Soluble biglycan also promoted autophagy in human peripheral blood macrophages. In experimental renal I/R, soluble biglycan induces the recruitment of M1 macrophages in a CD14 and CD44 dependent manner. Furthermore, circulating biglycan evokes autophagy in M1 macrophages while concurrently promoting M2 macrophage polarization and tubular repair, both of which are CD44 dependent. Using macrophages from mice lacking TLR2 and/or TLR4, CD14, or CD44, the authors demonstrated that the pro-autophagy signal required TLR4 interaction with CD44. Consequently, the biglycan-CD44 interaction drives M1 macrophage autophagy at 4 h post-renal I/R with further consequences on M2 polarization, tubular repair and regeneration at 7 days post-renal I/R [147]. This study shed light on the mechanism by which co-receptor dependent TLR4 mediated tubular recovery.

## 5. Conclusions

Albeit extensive research and notable progress in recent years, no specific and/or effective treatment of acute kidney injury has been found. TLR4 is characterized to have a central role in the activation of innate immune response recognizing both, pathogens and damage-associated molecular patterns. There are potential preclinical evidences that TLR4 has a deleterious role in acute kidney injury as it triggers an inflammatory and dysfunctional immune response that exacerbates tissue injury. With controlled inflammatory signals, the kidney undergoes repair and recovers function; however, with unresolved inflammation, TLR4 activation persists, resulting in a cycle of chronic renal injury (Figure 4). In addition, its role in kidney regeneration by regulating the production of cytokines, cell proliferation and survival, has also been gradually developed, which reflects TLR4′s multifaceted roles. Therefore, modulation of TLR4 activation and their downstream-related signaling pathways may represent promising targets for the design of new therapeutic strategies against an amplified TLR4 inflammatory response. Forthcoming studies in humans are necessary to validate the potential favorable effect of TLR4 inhibition against AKI.

Renal necrotic cells associated with ischemia/reperfusion injury induce the release of endogenous ligands, DAMPS. These DAMPs bind to TLR4 to elicit an innate immune response by promoting the release of pro-inflammatory mediators and recruiting immune cells to infiltrate the tissue. The immune cells that participate in these processes include, APC, such as dendritic cells and macrophages, as well as T cells and neutrophils. DAMPs may also stimulate adaptive immunity and participate in tissue repair.

Infections of pathogenic bacteria or viruses cause release of PAMPs that bind to TLR4 on parenchymal and immune cells, and stimulate an innate immune response that is accompanied by inflammation, activation of adaptive immunity, and eventually processes to resolve the infection and allow for tissue repair.

With controlled inflammatory signals, the kidney undergoes repair and recovers function. However, unresolved inflammation results in renal fibrosis and chronic kidney disease.

## Figures and Tables

**Figure 1 ijms-24-01415-f001:**
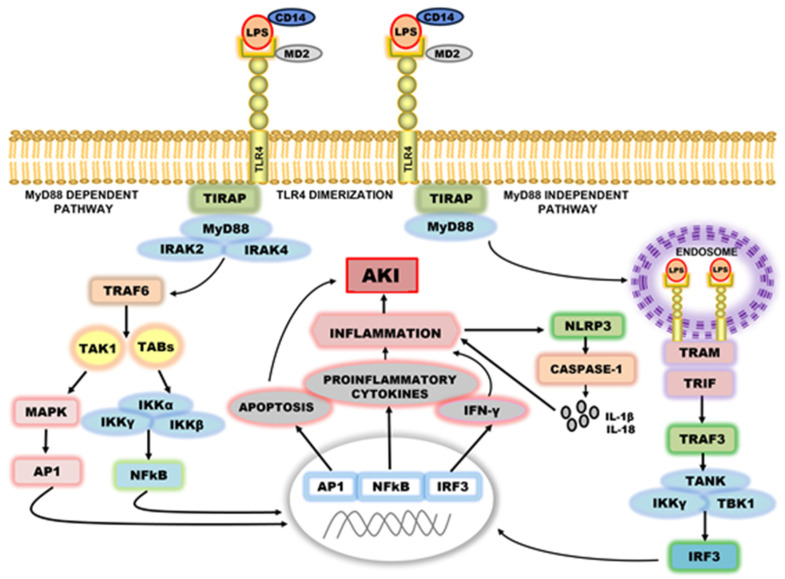
TLR4 signaling pathway. Abbreviations: AKI, acute kidney injury; AP-1, activating protein-1; CD14, cluster of differentiation 14; IFN-γ, interferon gamma; IKK, inhibitor of Kappa B kinase; IRAKs, interleukin-1 receptor associated kinase; IRF-3, interferon regulatory factor 3; LBP, lipopolysaccharides-binding protein; LPS, lipopolysaccharides; MAPKs, mitogen-activated protein kinase; MD2, myeloid differentiation factor 2; MyD88, myeloid differentiation primary response 88; NF-κB, nuclear factor kappa B; NLRP3, NLR family pyrin domain containing 3; TABs, TAK1-binding proteins; TANK, TRAF associated NF-κB activator; TBK1, TANK binding Kinase-1; TRAM, TRIF-related adaptor molecule; TIRA, TIR-domain containing adaptor protein; TLR4, toll-like receptor-4; TNF-α, tumor necrosis factor alpha; TRIF, TIR-domain-containing adaptor-inducing interferon-β.

**Figure 2 ijms-24-01415-f002:**
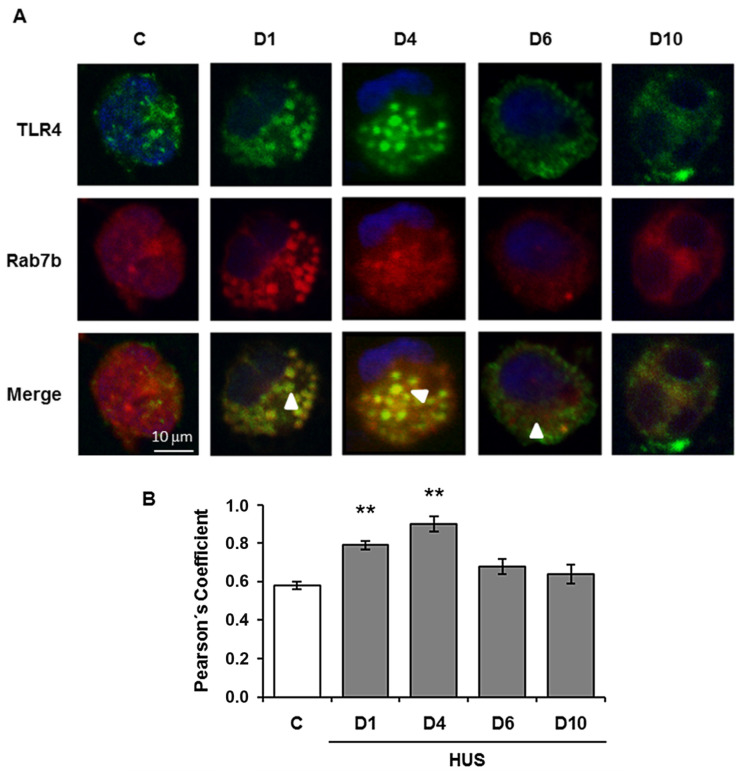
Colocalization analysis of Rab7b and TLR4 in monocytes during the acute course of STEC-HUS. TLR4 and Rab7b colocalization in monocytes was determined throughout the progression of HUS and in controls by indirect immunofluorescence and confocal microscopy. Specific rabbit anti TLR4/anti rabbit FITC (green), mouse anti Rab7b/anti mouse Cy3 (red), and Hoechst to visualize the nuclei (blue) were used. (**A**) representative immunofluorescence of both proteins in monocytes from one STEC-HUS patient and one control. The arrows indicate the colocalization areas. Scale bar 10 μm. Magnification 600×. (**B**). Graphic bar shows the TLR4 and Rab7b colocalization, which was determined by using the Pearson’s Coefficient. ** *p* < 0.01 vs. control. 20 cells from 3 independent cultures were analyzed for each day of the STEC-HUS follow-up. In addition, 20 cells from 5 independent cultures in the control group were evaluated. Data are expressed as the mean ± SEM. (Adapted from reference [122]).

**Figure 3 ijms-24-01415-f003:**
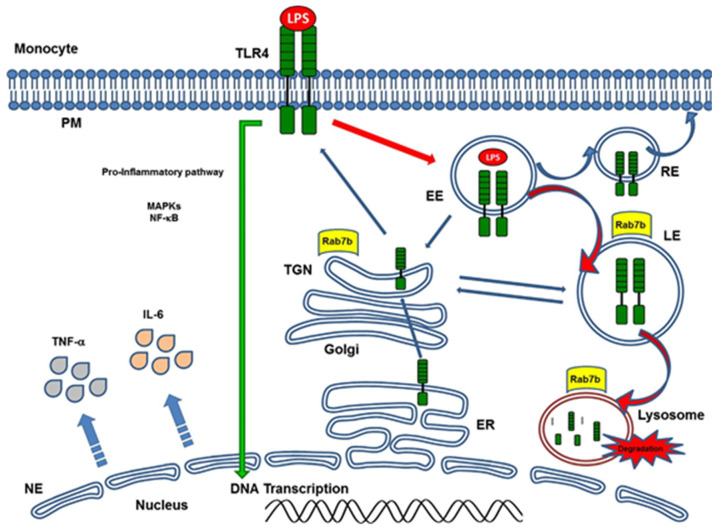
Involvement of Rab7b on the TLR4 (Toll-like receptor) endocytic pathway degradation. On the left, the inflammatory response to LPS/TLR4 internalization and TLR4 activation inducing enhanced intracellular cytokine generation on the very beginning of STEC-HUS (day 1 to 4), is presented. On the right, the figure displays the involvement of Rab7b on TLR4 degradation pathway leading to the downregulation of proinflammatory state in the early follow up of the disease (10 days). Abbreviations: LPS: lipopolysaccharide, PM: plasma membrane, NE: nuclear envelope, EE: early endosome, RE: recycling endosome, LE: late endosome, ER: endoplasmic reticulus, TGN: trans-Golgi network, DNA: desoxyribonucleic acid.

**Figure 4 ijms-24-01415-f004:**
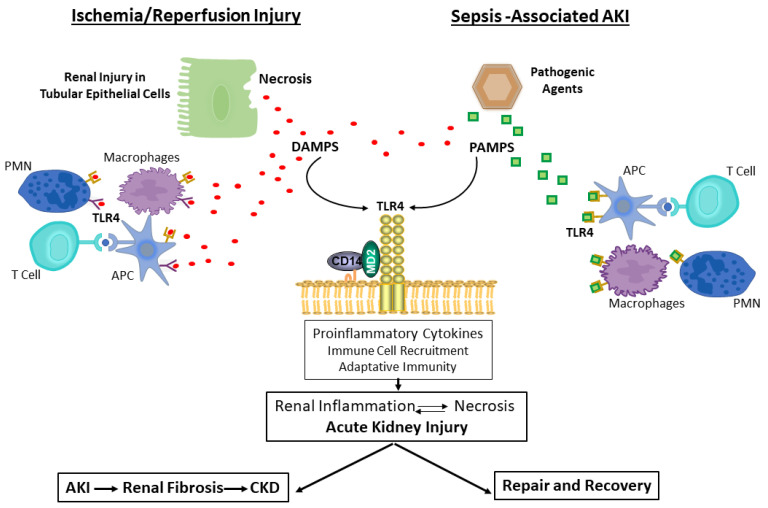
Role of TLR4 in the cycle of renal injury. Abbreviations: DAMPs, danger-associated molecular patterns; PAMs, pathogen-associated molecular patterns; APC, antigen-presenting cells; AKI, acute kidney injury; CKD, chronic kidney disease.

## Data Availability

Not applicable.

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
