# Peer review of "Toll-like Receptor 4 in Acute Kidney Injury"

_ijms, 2023, doi:10.3390/ijms24021415_

Round 1

Reviewer 1 Report

This is a good review that helps us better understand the role of TLR4 and its ligands activation in the inflammatory response to ischemic/reperfusion kidney injury and sepsis-associated AKI and promote us to find better therapeutic strategies against an excessive TLR4 inflammatory response. but I did not find the figure in the manuscript.

Author Response

Thank you very much for your comments. In the first version we uploaded to the web site the manuscript

separately from the figures, as it was required by the Editorial Manage

Reviewer 2 Report

the current manuscript is an excellent review in addressing the  role of TLR4, its endogenous ligands and activation in the inflammatory response to ischemic/reperfusion kidney injury and sepsis-associated AKI in addition to the potential regeneration signaling patterns of TLR4 in acute kidney injury. 

1- I did not see any figures in manuscript PDF but the text showing figure legends 1 to 3

2- I would highly recommend to add one paragraph discussing how can we antagonizing TLR in relation to acute renal injury 

Author Response

Answer to the Comments and Suggestions of the Reviewer.

Thank you very much for your comments.

In first version, we uploaded to the web site the manuscript separately from the figures, as it was required

by the Editorial Manager.

In this version, we added a brief paragraph discussing how some compounds have been tested in

preclinical studies for their capacity to block TLR4-mediated cytokine production, and few of them have

reached the clinical trials.

Title: TLR4 Inhibition in Acute Kidney Injury. Manuscript Page 15

Reviewer 3 Report

This is the important topic to understand the role of TLR4 on the acute kidney injury. Despite the manuscript is well planned and written, there are still many grammatical errors that need to be corrected for publication. Authors used very difficult language to descript the inflammatory cascade and unnecessary information throughout the manuscript that all needs to be simplified for reader.

In Abstract:

-associated with considerable high morbidity and mortality.

-the innate immune system plays an important role as the first defense mechanism to response to tissue injury.

-please clarify or simplify this sentence: Increasingly recognized is that endogenous molecules generated during tissue injury and labeled as damage-associated molecular pattern molecules (DAMPs), also activate pattern recognition receptors similarly to PAMPs, thereby offering an understanding of sterile types of inflammation.

Introduction- TLR4 signaling in ischemia/reperfusion-induced AKI

-please correct this sentence: Outstanding advances in understanding the cellular and molecular aspects have occurred, yet little progress has been made in the translation of these findings to the clinical area.

-the major factors in regulating inflammation are necrotic and apoptotic cells…. What are those major factors?

TLR4 signaling in sepsis associated AKI

-          Clarify this sentences… Early supportive interventions in sepsis reduce mortality, although it is less clear that they prevent or ameliorate sepsis-associated AKI, since specific mechanisms underlying AKI attributable to sepsis are not fully understood….

-           

After bacterial infection, a hyperactive dysregulated innate immune response leads to a cascade of proinflammatory molecules activating the complement system, and cellular innate immunity that contributes to sepsis-associated AKI

-          These studies show local microvascular dysfunction and patchy areas of oxygen

saturation, likely related to endothelial dysfunction and inflammatory and other oxidative factors [104]. But you addressed in only one reference…

TLR4 AND CELL REGENERATION

-The sentences are very difficult to understand….Kidney repair process includes interaction of complex events. Reassembly of the actin cytoskeleton, and repolarization of the surface membranes initiate the recovery of proximal tubular cells The renal response of repair is replacement of injured tubular epithelium with functional tubular cells but it happens in a restricted manner as kidney has a limited capacity to undergo endogenous tissue remodeling. During renal development mesenchymal to epithelial transition (MET) follows and this process is controlled by different growth factors …..

-It is the effector phenotype of the recruited macrophages rather than their presence alone that determines the extent of renal parenchymal injury as has been suggested in recent studies of other forms of immune- mediated renal injury. It has been proposed that inflammatory cells play a negative role.

- To limit overshooting immunopathology in sterile tissue injuries and allow tissue recovery, a number of counter regulatory mechanisms exist that mostly limit immune activation of intrarenal dendritic cells

-Although, surviving tubular epithelial cells (TECs) enter the cell cycle within few hours on injury, a functional tubular recovery does not occur before the resolution of sterile inflammation has occurred and the tubulointerstitial microenvironments become dominated by proregeneratory factors

Author Response

Thank you very much for your comments.

Answer: In relation to the inflammatory cascade, the description of the TLR4 signaling pathway is

included in Figure 1

In Abstract:

-associated with considerable high morbidity and mortality.

-the innate immune system plays an important role as the first defense mechanism to response to tissue

injury.

Answer: These corrections have been made.

-please clarify or simplify this sentence: Increasingly recognized is that endogenous molecules generated

during tissue injury and labeled as damage-associated molecular pattern molecules (DAMPs), also activate

pattern recognition receptors similarly to PAMPs, thereby offering an understanding of sterile types of

inflammation.

Answer: We have simplified this sentence.

¨Endogenous molecules generated during tissue injury and labeled as damage-associated

molecular pattern molecules (DAMPs), also activate pattern recognition receptors, thereby offering

an understanding of sterile types of inflammation¨.

-please correct this sentence: Outstanding advances in understanding the cellular and molecular aspects

have occurred, yet little progress has been made in the translation of these findings to the clinical area.

Answer: This sentence was corrected.

¨Advances in understanding the cellular and molecular aspects have been made, however small

progress has been achieved in the translation of these findings to the clinical area¨.

-the major factors in regulating inflammation are necrotic and apoptotic cells…. What are those major

factors?

Answer: This sentence was modified.

¨Tubular cell necrosis and apoptosis are major factors in the regulation of inflammation, since dying

cells release intracellular molecules which are referred to as DAMPs. DAMPs activate a set of

pattern recognition receptors on tissue-resident cells¨.3

TLR4 signaling in sepsis associated AKI.

-Clarify this sentences … Early supportive interventions in sepsis reduce mortality, although it is less clear

that they prevent or ameliorate sepsis-associated AKI, since specific mechanisms underlying AKI

attributable to sepsis are not fully understood….

Answer: I do not think this paragraph needs to be clarified. I apologize for this.

After bacterial infection, a hyperactive dysregulated innate immune response leads to a cascade of

proinflammatory molecules activating the complement system, and cellular innate immunity that contributes

to sepsis-associated AKI

Answer: This sentence was modified.

¨After bacterial infection, a hyperactive dysregulated innate immune response leads to a cascade

of proinflammatory molecules, contributing to sepsis-associated AKI by the activation of cellular

innate immunity and the complement system¨.

These studies show local microvascular dysfunction and patchy areas of oxygen saturation, likely related

to endothelial dysfunction and inflammatory and other oxidative factors [104]. But you addressed in only

one reference…

Answer:

¨These studies show local microvascular dysfunction and patchy areas of oxygen saturation, likely

related to endothelial dysfunction and inflammatory and other oxidative factors [103,104].

References;

103- Sun, N.; Zheng, S.; Rosin, D. L.; Poudel, N.; Yao, J.; Perry, H. M.; Cao, R.; Okusa, M. D.; Hu, S. Development

of a Photoacoustic Microscopy Technique to Assess Peritubular Capillary Function and Oxygen Metabolism in

the Mouse Kidney. Kidney Int 2021, 100 (3), 613–620. https://doi.org/10.1016/J.KINT.2021.06.018.

104- Zafrani, L.; Payen, D.; Azoulay, E.; Ince, C. The Microcirculation of the Septic Kidney. Semin Nephrol 2015,

35 (1), 75–84. https://doi.org/10.1016/J.SEMNEPHROL.2015.01.008

TLR4 AND CELL REGENERATION

-The sentences are very difficult to understand …Kidney repair process includes interaction of complex

events. Reassembly of the actin cytoskeleton, and repolarization of the surface membranes initiate the

recovery of proximal tubular cells The renal response of repair is replacement of injured tubular epithelium

with functional tubular cells but it happens in a restricted manner as kidney has a limited capacity to undergo 4

endogenous tissue remodeling. During renal development mesenchymal to epithelial transition (MET)

follows and this process is controlled by different growth factors ...

Answer: This paragraph was clarified.

Kidney repair process includes interaction of complex events. Reassembly of the actin

cytoskeleton, and repolarization of the surface membranes initiate the recovery of proximal tubular

cells.

The replacement of injured tubular epithelium with functional tubular cells happens in a restricted

manner, as kidney has a limited capacity to undergo endogenous tissue remodeling.

During renal development the mesenchymal to epithelial transition (MET) process is controlled by

different growth factors e.g. hepatocyte growth factor (HGF) and bone morphogenetic protein-7

(BMP-7).

-It is the effector phenotype of the recruited macrophages rather than their presence alone that determines

the extent of renal parenchymal injury as has been suggested in recent studies of other forms of immune

mediated renal injury. It has been proposed that inflammatory cells play a negative role.

Answer: This paragraph was clarified in the next paragraph.

The phenotypic switching of intrarenal mononuclear phagocytes away from classically activated

(proinflammatory) to alternatively activated (anti-inflammatory/proregeneratory) cells is necessary

for recovery on AKI.

-To limit overshooting immunopathology in sterile tissue injuries and allow tissue recovery, a number of

counter regulatory mechanisms exist that mostly limit immune activation of intrarenal dendritic cells.

Answer: This sentence was not changed.

-Although, surviving tubular epithelial cells (TECs) enter the cell cycle within few hours on injury, a functional

tubular recovery does not occur before the resolution of sterile inflammation has occurred and the

tubulointerstitial microenvironments become dominated by proregeneratory factors

Answer: This sentence was modified.

¨Although, surviving tubular epithelial cells (TECs) enter the cell cycle within few hours on injury, a

functional tubular recovery does not occur before the resolution of sterile inflammation is resolved

and the tubulointerstitial microenvironment become dominated by proregeneratory factors¨.

Note: Revisions to the manuscript have been marked up using the “Track Changes”

function.